# Structural coordinates: A novel approach to predict protein backbone conformation

**Vladislava Milchevskaya** [1]*, **Alexei M. Nikitin**[2], **Sergey A. Lukshin**[2], **Ivan V. Filatov**[3], **Yuri V. Kravatsky**[2], **Vladimir G. Tumanyan**[2], **Natalia G. Esipova**[2], **Yury V. Milchevskiy**[2]*

**1** Institute of Medical Statistics and Bioinformatics, Faculty of Medicine, University of Cologne, Cologne, Germany, **2** Engelhardt Institute of Molecular Biology, Moscow, Russia, **3** Moscow Institute of Physics and Technology, Dolgoprudny, Russia

* vmilchev@uni-koeln.de (VM); milch@eimb.ru (YVM)

**Data Availability Statement:** All relevant data are within the manuscript, its Supporting information files and our open repository http://pbpred.eimb.ru/S/index.html, which is also referenced in the text of the manuscript.

## Abstract

### Motivation

Local protein structure is usually described via classifying each peptide to a unique class from a set of pre-defined structures. These classifications may differ in the number of structural classes, the length of peptides, or class attribution criteria. Most methods that predict the local structure of a protein from its sequence first rely on some classification and only then proceed to the 3D conformation assessment. However, most classification methods rely on homologous proteins' existence, unavoidably lose information by attributing a peptide to a single class or suffer from a suboptimal choice of the representative classes.

### Results

To alleviate the above challenges, we propose a method that constructs a peptide's structural representation from the sequence, reflecting its similarity to several basic representative structures. For 5-mer peptides and 16 representative structures, we achieved the Q16 classification accuracy of 67.9%, which is higher than what is currently reported in the literature. Our prediction method does not utilize information about protein homologues but relies only on the amino acids' physicochemical properties and the resolved structures' statistics. We also show that the 3D coordinates of a peptide can be uniquely recovered from its structural coordinates, and show the required conditions under various geometric constraints.

## Introduction

Due to fast development of machine learning and statistical inference methods, recent years brought significant progress in protein local structure classification and prediction [1, 2]. The predicted structure of local protein fragments serves as a critical component in most global protein folding models, along with homology information and physicochemical properties of amino acids [3].

**Funding:** VM has been partially supported by the DFG FOR 2715. YM, AN, YK, VT and NE have been partially supported by the Russian Foundation for Basic Research grant 20-04-01085.

**Competing interests:** The authors have declared that no competing interests exist.

In the context of local structure prediction, the ultimate goal is to reconstruct the atomic coordinates of a protein fragment as precise as possible, given its amino acid sequence. A lot of information can be drawn from the known structures of the fragments with similar characteristics. For that reason, many local structure prediction methods rely on these fragments' classification into certain structural classes. These can be secondary structure-based classifications, as well as classifications based on so-called structural alphabets or protein blocks (PBs) [4].

Secondary structure assignments predominantly exploit hydrogen bond information to group protein fragments. One of the widely used such methods, DSSP, clusters protein 5-mers into eight classes: Seven of the groups reflect certain structural patterns, such as $\alpha$-helices, $\beta$-sheets, 3/10 helices et.c., and the eighth "blank" group is used if no other structural pattern applies [5]. In another structural assignment method STRIDE, Frischman et al. extend the DSSP definition of individual secondary structures: Apart from the hydrogen bonds, it includes criteria based the dihedral angle frequencies [6]. The DSSP and STRIDE assignments yield interpretable structural classes, and methods such as PSIPRED and SSpro8 predict secondary structure from the protein sequence with high accuracy [7, 8].

On the other hand, a structural alphabet may suffer in terms of interpretability, but it does not usually contain a "blank" undefined class. The aim here is to cover the full range of known conformations with a relatively small set of representative structures. There are various approaches to do so: Some exploit the distance matrix between $C_\alpha$ atoms of the backbone fragment and use the cumulative distance between the corresponding matrix elements to measure dissimilarity between two protein fragments [9, 10]. Sander et al, for instance, proposed a structural alphabet containing 27 clusters of 7-mer fragments [9]. Using a slightly different approach, Hahn et al. first cluster fragments based on their sequence similarity, and then examine structural variation within clusters [11]. The ones with high within-cluster variation are discarded, while others are kept to represent sequence neighbourhoods. One of the most used structural alphabets at the moment has been developed by de Brevern et al. [12] There, the authors clustered all non-redundant five-mer peptides available in the PDB at that time into 16 groups based on the RMSDA (root mean square deviation of angle) distance between the peptides. The authors refer to this alphabet as protein blocks (PB).

Joseph et al. review various applications of the Protein Blocks (PB): Multiclass local protein structure prediction (LOCUSTRA), structural alignment (PB-ALIGN and iPBA), identification of functional structural motifs in proteins et.c. [2, 13–15].

These and most other methods for local structure prediction available in the literature are multiclass predictions: Namely, a protein fragment is assigned to the most similar protein block. However, there is significant variation in structure within some clusters: Even if correctly classified, two protein fragments may have the same cluster assignment but be notably different in structure. This may hamper the use of Protein Blocks or other structural alphabets when atomic coordinates of a protein fragment need to be reconstructed with high resolution.

Here we propose a new approach for using alphabets of basic structures, which aids at recovering the 3D coordinates of a protein fragment. By representing a query fragment in terms of distances to each of the basic structures in the alphabet structures, we retain the information needed to reconstruct the 3D coordinates unambiguously. We also show that this approach yields better multiclass prediction accuracy than the current gold-standard. Additionally, we present a simple lower bound for the number of basic structures needed for unambiguous coordinate reconstruction, which depends on the length of the protein fragment and whether certain bond length, bond angles and dihedral angles are assumed fixed.

## Results

### Relation of RMSD- and RMSDA-based assignments

For local structure classification and prediction, one of the most used sets of protein structures is a set of the 16 protein blocks (PBs) presented by De Brevern et al. [12]. These PBs serve as cluster centres: A given protein structure is classified according to its nearest protein block and assigned the corresponding cluster label. In the original article, the authors use the RMSDA (root mean square deviation of angular values) between the corresponding backbone atoms to define the distance between two protein fragments of the same length.

Protein Blocks are five-residue long structural fragments defined through their dihedral angles. Alternatively, one can represent PBs with their 3D coordinates under a common assumption that the bond angles and bond lengths have their standard values, and the omega angle is 180 degrees.

Here we use the same PBs as cluster centres, but utilise their representation via 3D coordinates. The assignment is then based on the RMSD distance: Namely, a query protein fragment is assigned the labels of its RMSD-closest protein block. We then compared the RMSD-based assignment (referred to in capital letters) with the assignment calculated using the RMSDA (lower case letters), utilising a set of protein fragments from the PDB30 (further referred to as *training-PDB30* and described in Methods). Remarkably, the two assignments agree only for 69.6% of the 5-residue fragments.

To investigate the assignments' difference, we first assessed if the coarse characterisation of a cluster depends on the distance choice. For that, we used the characterisation used in the original article. As expected, the RMSD-clusters' coarse characterisation agrees with the one presented by de Brevern et al.: Namely, if a cluster consists of N-cap $\beta$ structures in its RMSDA composition, the same holds for the RMSD composition (see Table 1). Similarly, if a cluster consists mainly of coils, it will hold for both RMSD- and RMSDA- based compositions.

Further, we investigated each cluster separately. For the clusters that describe the major regular structures, such as $\alpha$–helices (*M*) and $\beta$–strands (*D*), the clusters' relative compositions differ only slightly between RMSDA and RMSD (Fig 1A–1C): The RMSD-assignment of the fragments agrees with their RMSDA labels by 97,4% and 88% correspondingly. However, one may note that this relation is one-sided, i.e. if a fragment was classified as *M* or *D* using RMSD, it will most likely be assigned *m* or *d* respectively by RMSDA. The reserve is not true: Far not every *m* fragment (RMSDA) will be attributed to cluster *M* by RMSD (Fig 1C). We believe this asymmetry is due to the following fact: A five-residue fragment has 8 dihedral angles; a change in the central ones may cause a larger RMSD between the two fragments, while the same change in one of the flanking angles would influence the distance less. In both cases, the RMSDA between the fragments would be the same. This consideration would be even more valid for non-regular structures further.

For the clusters that describe less regular structures but relatively similar to those above (*K*, *L*, *N* being the closest to the $\alpha$– helices, and *C*, *E*, *F* to the $\beta$–strands), the agreement between RMSDA- and RMSD-assignments is weaker (Fig 1C). However, most of the varying classifications for these clusters usually fall within the group of $\beta$-sheet-like (*C*, *D*, *E*, *F*) or $\alpha$-helix-like (*K*, *L*, *M*, *N*) structures. We have also observed that the RMSD between the central structures $PB_c$, $PB_d$ and $PB_e$ is small, explaining some of the assignment discrepancies within the beta-sheet group (S6 Table and S5 Fig in S1 File). Similarly, the RMSD between $PB_m$ and $PB_n$ explains why some of the *m*–fragments from the RMSDA assignment are classified as *N* by RMSD.

Finally, the most considerable discrepancy between RMSDA- and RMSD- assignments appears for clusters that cover irregular structures: (*G*, *H*, *I*, *J*). Fig 1A shows that for the *G*-

**Table 1. Description of protein clusters in terms of RMSD.**

| PB | RMSDA PB freq % | RMSD | | | | coarse characterization |
|---|---|---|---|---|---|---|
| | | PB freq % | mean Å | sd Å | neighbours (ordered by distance) | |
| A | 3.68 | 5.34 | 0.85 | 0.78 | gcpfedbihkljomn | N-cap $\beta$ |
| B | 4.29 | 5.70 | 0.91 | 0.90 | phfkeagcdiljmno | N-cap $\beta$ |
| C | 8.31 | 4.52 | 0.81 | 0.72 | de fagpbkhilmjno | N-cap $\beta$ |
| D | 18.68 | 12.84 | 0.62 | 0.45 | ce fagbphkilmjno | $\beta$ |
| E | 2.18 | 3.75 | 0.83 | 0.75 | fdc abgphkjilmno | C-cap $\beta$ |
| F | 6.45 | 5.79 | 0.83 | 0.75 | e dcabgphkjimlno | C-cap $\beta$ |
| G | 1.14 | 6.01 | 1.00 | 1.04 | acpfebidohjklmn | mainly coil |
| H | 2.1 | 1.33 | 0.90 | 0.89 | kj bpogfinamlecd | mainly coil |
| I | 1.49 | 3.83 | 0.95 | 0.97 | oplnmjgakbhfced | mainly coil |
| J | 0.83 | 1.73 | 0.90 | 0.87 | h koipgbnamlfecd | mainly coil |
| K | 5.25 | 5.33 | 0.69 | 0.53 | h jbpnmliofgaecd | N-cap $\alpha$ |
| L | 5.2 | 4.65 | 0.59 | 0.45 | mn ikobgpahjfced | N-cap $\alpha$ |
| M | 32.98 | 30.26 | 0.24 | 0.12 | ln ikobhgjpafced | $\alpha$ |
| N | 1.78 | 3.38 | 0.63 | 0.51 | ml ikojhbgpafced | C-cap $\alpha$ |
| O | 2.44 | 2.16 | 0.68 | 0.58 | ijnghlmkapbfced | C-cap $\alpha$ |
| P | 3.29 | 3.37 | 0.83 | 0.81 | baigkchfjedlomn | C-cap $\alpha$ to N-cap $\beta$ |

The 'PB' column states the name of the protein block in accordance with de Brevern et al. [12]; 'PB freq' is the relative frequency of the instances of a corresponding cluster among all protein fragments, the clusters are composed of the nearest neighbours to the corresponding protein block in terms of RMSDA or RMSD. The 'mean' and 'sd' columns show the mean and the standard deviation of the RMSD of the corresponding clusters, in angstroms (Å). The 'neighbours' column shows the nearest Protein Blocks to the corresponding PB, in terms of RMSD. The coarse characterization corresponds to the classical secondary structure.

fragments that deviate from $PB_g$ by 0.5Å or more, it is unclear to which of the RMSDA-clusters it can be assigned, both regular or irregular. A similar assignment discrepancy can be observed for the remaining irregular clusters *I*, *J* and *E*. In comparison, the distance between almost any two cluster centres (PBs) is larger than 1Å(see S6 Table in S1 File).

While RMSDA- and RMSD-assignments agree well for regular structures and compact clusters (such as *m* or *d*), the less regular the structure, the higher the discrepancy between the two assignment systems is. In the context of the clusters that cover irregular structures, the two distances have a qualitatively different meaning, and the choice of the distance has to be considered with care. Since further we will aim at the coordinate reconstruction rather than fragment classification, we will need to approximate distances between atoms. For this purpose, we will utilise RMSD.

## RMSD recovers ambiguity of RMSDA

Dihedral angles of a protein fragment represent one of the most important characteristic of the structure, and researchers often rely on this characteristics to classify structures. For instance, the standard alpha-helical fragment of five residues would have the following dihedral angles: $\phi_i = -57$, $\psi_i = -47$, $i = 1, \ldots, 5$. Therefore, it is only logical to use an angle-based distance (RMSDA) between protein fragments to assess their similarity.

However, we have noticed that two protein fragments may have the same RMSDA to the standard helical structure but their actual alignment in 3D may differ significantly. Fig 2 shows an example to illustrate so: Four protein fragments of 5 residues (in green) aligned to the standard helical fragment (red). For the fragments in (A)-(C), RMSDA of the green and the red structures is the same (14 degrees is a relatively small value of RMSDA for both data sets we

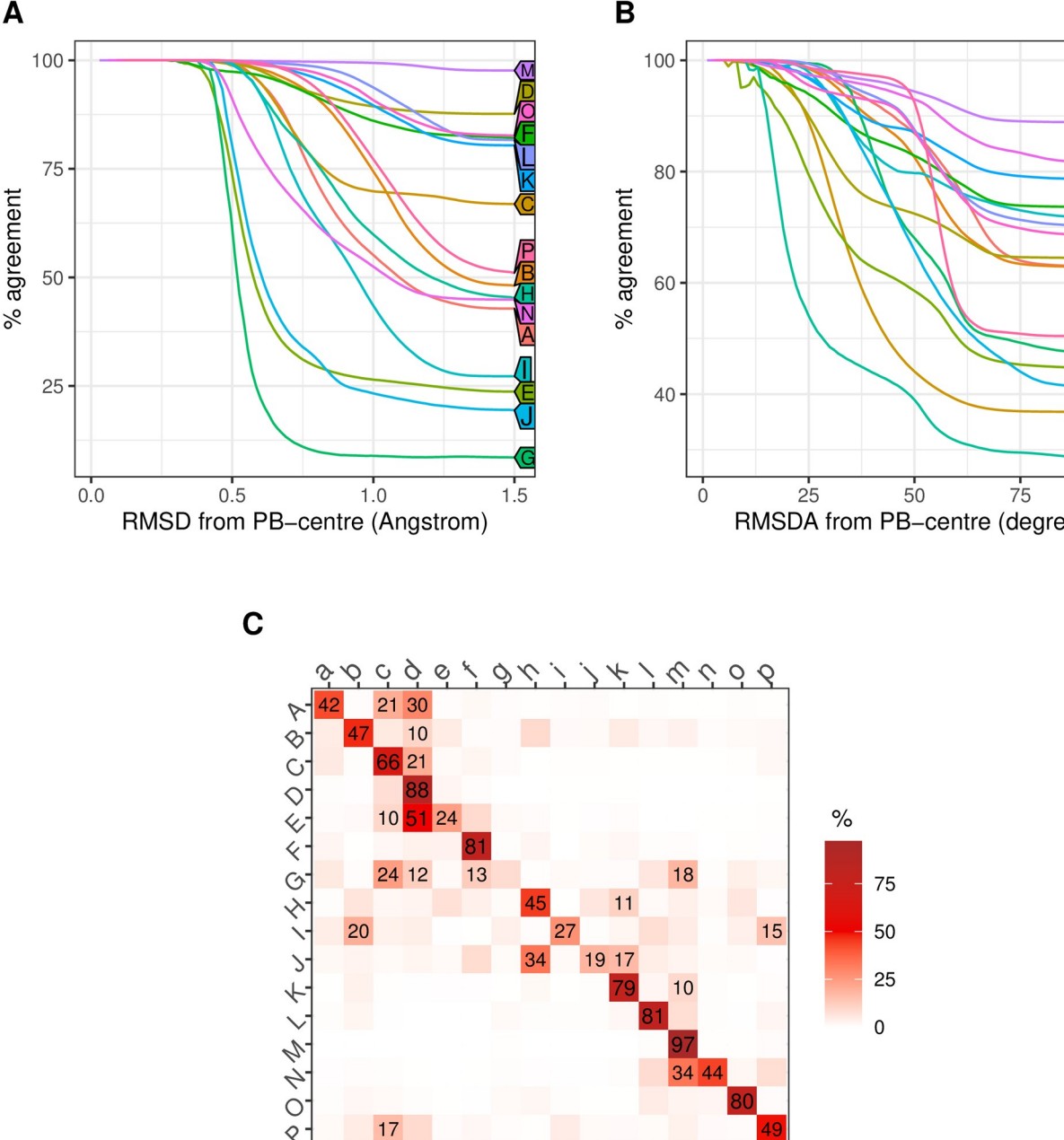

**Fig 1. Relation between assignments based on RMSDA and RMSD.** A shows the percentage (Y-axis) of matching RMSDA- and RMSD assignments among all the RMSD assignments for a given cluster that lie within a certain distance (X-axis) from the cluster centre, i.e. the corresponding protein block (PB); B shows the reverse of A, namely, the percentage of matching RMSDA- and RMSD assignments among all the RMSDA assignments for a given cluster that lie within a certain RMSDA-distance (X-axis) from the centre; C shows the percentage of matching assignment for each RMSD-cluster among all fragments. The full set of fragments for this investigation is *PDB30* (see Methods for details).

investigated, see Methods for details). Yet, the 3D dissimilarity between the standard helix and the first fragment is very small, whereas the second and the third fragments are noticeably different from the standard alpha-helix. The fourth fragment represents a rare case of the peptide bond angle $\omega \neq 180$: in terms or RMSDA it is identical to the alpha-helix, while the two structures are considerably different.

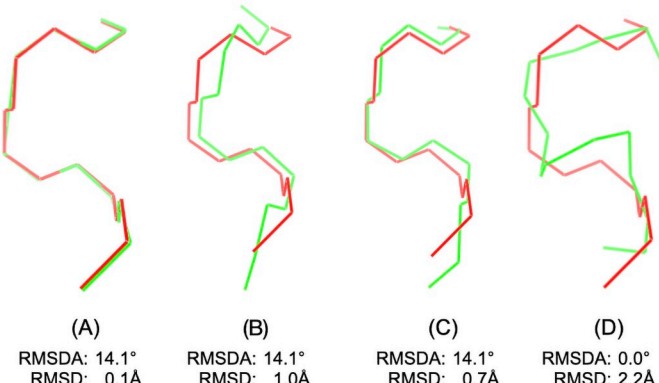

**Fig 2. Discrepancy examples between RMSDA and RMSD.** Structural alignment of a 5-residue protein fragments (green) against the standard alpha-helical structure (red) with dihedral angles $\phi_i = -57$, $\psi_i = -47$, $i = 1, \ldots, 5$. The examples (A)-(C) are chosen such that to highlight that one can find a (green) fragment with the same RMSDA distance value to the reference (red) fragment but varying RMSD. In (D), on the contrary, we show an example of a (green) fragment that has RMSDA of 0 to the reference (red), but a large RMSD. Dihedral angles of the fragment (A) are $\phi_i = -61.1$, $\psi_i = -42.8$, $i = 1, \ldots, 5$, its RMSDA to the standard helix is $14.1°$ and its RMSD to the standard helix is 0.09 Å; Dihedral angles of the fragment (B) are $\phi_i = -32.9$, $\psi_i = -42.9$, $i = 1, \ldots, 5$, its RMSDA to the standard helix is 14.1421 and its RMSD to the standard helix is 1.0065; Dihedral angles of the fragment (C) are all $\phi_i = -57$, $\psi_i = -47$, except for $\phi_3 = -17$, its RMSDA to the standard helix is again 14.1421 and its RMSD to the standard helix is 0.74. Dihedral angles of the fragment (D) are all $\phi_i = -57$, $\psi_i = -47$, except $\omega_3 = 0$ instead of the usual 180.

These examples illustrate that structural similarity between protein fragments is not captured by the RMSDA measure entirely. Another situation in which RMSDA may under- or overestimate the similarity between two three-dimensional fragment may be, for instance, if all their dihedral angles are identical except for one angle in the middle (for instance, $\psi_3$ in a 5-residue fragment): These structures will not align close, even though their RMSDA may be small. In short: Angle values in the middle of a fragment have higher impact on its 3D alignment to the reference fragment, as compared to the flanking angles.

In contrast, RMSD is prone to such position-specific effect due to the way it is defined: The two protein fragment are first aligned against one another to find the relative position in which they are most similar (in terms of RMSD), which makes it more robust against the position-specific effect we observed for RMSDA. Based on these considerations, we chose to use RMSD as distance measure between protein fragments over RMSDA proposed in the original article [12].

## Reconstruction of the backbone conformation

Here we aim to show that distances to PBs can be used as "alternative coordinates", from which one can unambiguously reconstruct the conformations of a protein. There, a five-residue fragment with a known structure is represented by 16 numbers: The RMSDs between the fragment and each of the PBs. Let the distances to the protein blocks $PB_a$, $PB_b$, ..., $PB_p$ be denoted as $(\tilde{R}_1, \tilde{R}_2, \ldots \tilde{R}_{16})$ correspondingly.

If we now forget the structure of the original fragment but retain only $(\tilde{R}_1, \tilde{R}_2, \ldots \tilde{R}_{16})$, would this information be enough to reconstruct the structure of the original fragment? To investigate this, we proposed an optimisation scheme similar to the one in Molecular Dynamics but with a different loss function.

Let us fix a conformation of a five-residue fragment $x$ using its dihedral angles:

$$\Psi_x = \{\psi_1, \phi_2, \psi_2, \phi_3, \psi_3, \phi_4, \psi_4, \phi_5\},$$

For the proposed fragment $x$, we can also calculate RMSDs to the protein blocks (let the notation for them be $R_i = R_i(\Psi)$, $i = 1, \ldots, 16$). If the structures of the original fragment and the proposed fragment $x$ are similar, the 16 distances should be the approximately the same, i.e.

$$\tilde{R}_i \approx R_i(\Psi), \quad i = 1, \ldots, 16$$

However, if the distances are the same, it does not guarantee that the original fragment and fragment $x$ have similar structures. To check that, we attempted to reconstruct an exemplary structure's dihedral angles, its actual values shown in Table 2. For that, we minimised the squared loss as a function of angles

$$L(\Psi_y) = \sum_{i=1}^{16}(\tilde{R}_i - R_i(\Psi_y))^2,$$

using two optimization methods: The method of conjugate gradients and the Fletcher-Powell algorithm, the starting point for both algorithms was set to $\Psi_0 = (0, 0, 0, 0, 0, 0, 0, 0)$ [16, 17]. Table 2 shows that both methods reached the true values of dihedral angles with high accuracy. This example allows to conclude, that such alternative coordinates, at least in this exemplary case, retain the information needed to reconstruct the structure of a protein fragment.

## Number of PB needed for reconstruction

We have seen that with 16 basic structures (protein blocks), it is possible to reconstruct the backbone coordinates of a 5-residue protein fragment, given the bond lengths and most of the bond angles are fixed, and the omega angle is assumed 180 degrees. However, it is not always a valid assumption: We have assessed the *PISCES30* dataset and observed that approximately 0.27% of the residues take the cis conformation (omega of 0 degrees). This agrees with the 0.31% reported by Joseph et al. [18]. Consequently, we expect 0.27%*4 > 1% of five-residue fragments to have at least one residue in the cis conformation. For a longer protein fragment, such cis/trans conformations may be equilibrated. However, if not, disregarding the cis confirmation may cause a significant prediction error. Additionally, some omega angles deviate from 180 degrees in trans conformation: We observed more than 7% residues' omega in *PISCES30* to deviate from the straight angle by mo than 10 degrees. Altogether, these deviations from the ideal trans conformation may lead to a noticeable loss in precision.

**Table 2. Predicted and actual angles.**

|  | True angles | Conj. Grad. | Fletcher-Powell |
|---|---|---|---|
| $\psi_1$ | 108.2 | 108.0 | 108.2 |
| $\phi_2$ | -90.1 | -90.0 | -90.1 |
| $\psi_2$ | 119.5 | 119.5 | 119.5 |
| $\phi_3$ | -92.2 | -92.0 | -92.2 |
| $\psi_3$ | -18.0 | -18.4 | -18.1 |
| $\phi_4$ | -128.9 | -128.6 | -128.9 |
| $\psi_4$ | 147.0 | 147.4 | 147.0 |
| $\phi_5$ | -99.9 | -100.4 | -99.9 |

Predicted and actual dihedral angles of the protein fragment $x$. The 'True angles' are the actual dihedral angles of the protein fragment $x$; the 'Conjugate Gradients' and 'Fletcher-Powell' show the angle values reconstructed from the distances to the 16 protein blocks using the conjugate gradients and Fletcher-Powell algorithms correspondingly.

**Table 3. Degrees of freedom.**

| Geometric constraints | deg. of freedom |
|---|---|
| Positions of all 3$N$ atoms | 9$N$ |
| Rotation and Translation invariance | 9$N$ − 6 |
| Bond lengths are fixed | 6$N$ − 5 |
| Bond angles are fixed | 3$N$ − 3 |
| Peptide bond angles $\omega$ are fixed | 2$N$ − 2 |

The minimal number of degrees of freedom required to describe an N-residue protein backbone, given a set of geometric constraints.

Here we calculate the minimal number of basic structures required in the general case. Let us consider a N-residue fragment, each amino acid corresponds to three backbone atoms ($N$, $C_\alpha$, $C$), and each atom has three coordinates in 3D. Thus, to define the position of the backbone, one needs 9$N$ degrees of freedom. Further, if the backbone is shifted along $X$, $Y$ or $Z$ axis, the structure is unchanged—thus, one can omit the 3 translational degrees of freedom. Similarly, the three rotational degreed of freedom do not represent the protein structure, and can be omitted as well. In most proteins, the change in bond lengths can be neglected—therefore, one may subtract 3$N$ − 1 degrees of freedom. Fixing bond angles would subtract additional 3$N$ − 2 degreed, and setting the $\omega$ angle (peptide bond) to 180 would release another $N$ − 1 degrees of freedom. Table 3 sums up these considerations.

Therefore, for a 5-residue fragment with all bond length and bond angles considered fixed, one would need at least 12 (fairly unrelated) basic structures to reconstruct the conformation of the backbone. With the same restrictions for a 9-residue fragment, at least 24 basic structures would be required.

## Benchmarking Q16

Here we return to the classification problem and benchmark our method against the two leading tools for protein local structure prediction: LOCUSTRA and PB-kPred [12, 13]. In order to achieve comparable figures, we use the same protein fragment size (five residues) and the same set of proteins as in PB-kPred (*PDB30*) for our model training and testing.

S8 Table in S1 File represents the prediction accuracy calculated for each structural cluster. shows that for clusters *B, D, F, G, J, K, M, O* and *P* out method has higher prediction accuracy than the two other methods, whereas for clusters *A, C, H* and *N* PB-kPred shows a better performance. Overall, the accuracy of our Q16 classification is 67.9% using the *PDB30* data set which is 6% higher than the one reported for PB-kPred for the same dataset [19]. We have also included the number reported by LOCUSTRA, however, those were calculated for a dataset different from the *PDB30*, therefore, the reported numbers cannot be compared directly to our method or PB-kPREd, but rather bring an overview.

Importantly, unlike the two other prediction methods, our model does not rely on the information about homologous proteins. Therefore, the distribution of prediction accuracies for our method is a unimodal Gaussian, but not bimodal as for PB-kPred, where the high prediction accuracy corresponds to those proteins that have close homologs outside the training sample (see S1 File).

Further, we have used a slightly larger and more up-to-date dataset *PISCES30* to assess our method's classification accuracy. For that, we have partitioned the *PISCES30* into a training set (3/4 of the protein chains) and s test set (1/4 of the protein chains). The Q16 accuracy on this

data set was 72%, which is even higher than for the *PDB30*. We believe the increase in prediction performance, in this case, is due to *PISCES30* being composed only of proteins structures resolved by X-Ray. The previous dataset *PDB30* contains a fraction of proteins resolved by NMR, including membrane proteins. The physicochemical properties of those proteins differ drastically from those of non-membrane. However, our prediction model is based on the physicochemical properties and trained largely in non-membrane proteins, it is not suited for the membrane proteins.

## Benchmarking using Structural Words

Further, we assessed the RMSD- and RMSDA-based assignments' performance in the case of overlapping protein fragments. Namely, we borrowed the idea of Structural Words from de Brevern et al. [20]: There, the authors use the structural alphabet of PBs to represent the structure of peptides longer than five residues. For instance, a five-letter Structural Word *mnopa* would represent a fragment of nine residues, where the first five residue fragment is assigned to $PB_m$, the 2-6 residues fragment to $PB_n$, the 3-7 to $PB_p$ etc. Among other things, the researchers investigated which of the five-letter structural words are most common among the resolved structures. Further, they assessed the distribution of RMSD within each structural group of the top SWs.

We have reproduced the above procedure for two types of Structural Words: Those based on RMSDA-assignment, which corresponds to the original article, and those based on RMSD. For that, we chose a 1000 protein chains at random from PISCES30 (see Methods for the dataset description), assigned a lower-case Structural Word to each nine-residue fragment using RMSDA, and an upper-case Structural Word using RMSD. Table 4 shows that the most common SWs are almost identical for both assignments. This served as a quality check of the RMSD-based assignment.

Further, following the original article, we calculated RMSD between each pair of fragments assigned to the same SW group, using *N*, $C_\alpha$, *C* atoms. We observed that the RMSD assignment almost always yields a more compact structural cluster, the data for the top occurring SWs in Table 4. Additionally, we discovered that the RMSD assignment yields approximately 7000 SWs, while RMSDA less than 4000 (for this particular draw of 1000 protein chains, it was 7232 upper-case SWs and 3731 lower-case). To ensure that the above results are not affected by possible sampling biases, we repeated all the calculation presented in Table 4 and observed similar figures.

We, therefore, conclude that RMSD assignment brings in improvement both in terms of precision of the Structural Words (lower within-group variability) and has higher descriptive flexibility (a larger set of words).

## Materials and methods

### Structure database generation

**Datasets for training and testing.**   We use two different datasets throughout the paper, further referred to as *PDB30* and *PISCES30*.

*PDB30*. The PDB contains a high number of nearly identical molecules, which may add undesired confounding factors at the model training step. To alleviate this, we used a representative set of sequence-unique structures with high resolution. Furthermore, to ensure that our results are comparable to those from the PB-kPRED prediction method, we opted for the PBD30 dataset comprised by Vetrivel et al. [19]. The PBD30 dataset contains 15,544 protein chains restricted to 30% sequence identity according to BLASTclust characterisation [21]. Among the sequences that shared more than 30% identity, the researches chose one representative protein chain that corresponds to the best available structure. Out of the 15,544

**Table 4. Structural Words for RMSDA and RMSD assignments.**

| Structural Word | % fragments RMSDA as. | % fragments RMSD as. | av. RMSD for RMSDA | av. RMSD for RMSD | ratio |
|---|---|---|---|---|---|
| MMMMM | 17.691 | 15.834 | 0.729 | 0.512 | 1.423 |
| DDDDD | 4.036 | 2.124 | 1.746 | 1.231 | 1.418 |
| LMMMM | 2.440 | 2.205 | 0.942 | 0.741 | 1.271 |
| KLMMM | 2.313 | 2.130 | 1.239 | 0.940 | 1.318 |
| FKLMM | 1.965 | 1.330 | 1.589 | 1.139 | 1.395 |
| CDDDD | 1.754 | 0.857 | 1.783 | 1.314 | 1.356 |
| DDDDF | 1.399 | 0.722 | 1.826 | 1.376 | 1.327 |
| MMMMN | 1.268 | 1.703 | 0.801 | 0.716 | 1.118 |
| MMMNO | 1.237 | 1.047 | 1.020 | 0.742 | 1.374 |
| MMNOP | 1.117 | 0.981 | 1.301 | 1.025 | 1.269 |
| DDDFK | 1.108 | 0.397 | 1.911 | 1.360 | 1.405 |
| MNOPA | 1.063 | 0.652 | 1.632 | 1.186 | 1.376 |
| DDFKL | 0.959 | 0.212 | 1.978 | 1.329 | 1.488 |
| DFKLM | 0.956 | 0.240 | 1.933 | 1.227 | 1.575 |
| ACDDD | 0.916 | 0.763 | 1.738 | 1.387 | 1.253 |
| BDCDD | 0.861 | *** | 1.954 | *** | *** |
| FBDCD | 0.747 | *** | 2.159 | *** | *** |
| DDDFB | 0.735 | 0.450 | 1.940 | 1.427 | 1.359 |
| CCDDD | 0.724 | 0.077 | 1.874 | 1.358 | 1.379 |
| DCDDD | 0.662 | 0.057 | 1.917 | 1.207 | 1.588 |

The most common Structural Words for the PISCES30 dataset, obtained through RMSDA- and RMSD-based structural assignments. The SWs are sorted by the topmost common by the RMSDA assignment. The percentage of 9-residue fragments assigned to each SW class is shown in columns two and three: the second column for the RMSDA assignment and the third column for the RMSD. Columns four and five show the average distance between the backbone atoms $N$, $C_\alpha$, $C$ of the fragments assigned to the same SW, for each assignment respectively; the last column shows the ratio of the distances in columns four and five. The *** sing stands for the SWs that were not present in the RMSD-based assignment.

structures, over 14000 were crystallographic, approximately 1100 were NMR-resolved, and 209 were solved by electron microscopy. The full list of the PDB identifies can be found at http://pbpred.eimb.ru/S/LS.txt.

The training and testing sets were obtained by partitioning the full set of 15,544 protein fragments into five equally sized subsets. Further, four of these subsets are used for training, and the remaining subset is used for testing. We refer to them as *training-PDB30* and *test-PDB30*.

*PISCES30*. The PISCES dataset is an updated version of the PDB, culled by resolution and sequence identity [22]. We used the resolution cutoff of 2.5 Angstrom, the R-factor cutoff of 1.0 and the allowed for sequence identity of 30%. In total, the PISCES30 dataset contains 17148 protein chains, the structures of which are resolved by X-Ray (no NMR-resolved structures).

**Dissimilarity measure.** To classify each of the pentapeptides from the training set, we use the 16 protein blocks (PB) developed by de Brevern et al., also referred to as Q16 [12]. Since PBs have the same length of five residues, one can calculate RMSD (root mean square deviation) between a pentateptide and a protein block $PB_i$, $i = 1, \ldots, 16$.

The smaller the RMSD is, the closer the two protein fragments are in terms of structure. Further, we define a similarity measure:

$$D(V_1, V_2) = \frac{1}{\lambda + RMSD(V_1, V_2)}, \tag{1}$$

where $\lambda$ is a fixed constant such that $\lambda \geq 0$. Here large values of $D$ correspond to more structurally similar fragments. To assign a five-mer protein fragment to one of the Q16 classes, we calculate its similarity value to each of the 16 protein blocks $PB_i$, $i = 1, \ldots, 16$.

$$D_i(V_1) = D(V_1, PB_i), \quad i = 1, \ldots, 16,$$

and choose the class with the highest similarity value.

## Structural alignment

Computation of the RMSD between two protein fragments requires optimisation over all possible alignments of these fragments against each other. Kabsch et al. introduced an optimisation procedure and showed that it reached the minimum of the RMSD function [23, 24]. We adapted a version of this procedure implemented by Gans and Shalloway in their tool *Qmol*, and integrated it into our pipeline [25].

$$\text{RMSD}(V_1, V_2) = \min \sqrt{\frac{\sum_{i=1}^{3M} [x_i^1 - x_i^2]^2 + [y_i^1 - y_i^2]^2 + [z_i^1 - z_i^2]^2}{3M}} \tag{2}$$

*where M is the number of residues in a fragment (in the original article M = 5), $V_1$ and $V_2$ denote the fragments, and $x_i^1 = x_i(V_1), y_i^1 = y_i(V_1), z_i^1 = z_i(V_1), \quad i = 1, \ldots, 3M$ denote Cartesian coordinates of the backbone atoms N, $C_\alpha$, C of the M–residue fragment $V_1$, and the minimum is taken over all spatial co-localization of the two fragments $V_1$ and $V_2$.*

## Feature generation

Generation of a comprehensive pool of features is the main and most laborious step of the prediction algorithm.

The sequence of a protein fragment can be characterised by the properties of amino acids forming this fragment as well as its flanking regions. At the model selection step, we tried flanking regions of various length, including the complete protein sequence. Two main types of information were used to generate a predictor pool: Physico-chemical properties of amino acids and RMSD distribution of fragments with the identical sequence to each of the protein blocks $PB_i$, $i = 1, \ldots, 16$.

**RMSD-based predictors.** Reflect the distribution of distances from a given sequence fragment to the 16 protein blocks. These distributions are derived from the identical sequence instances in the training set. Unsurprisingly, we encountered the following problem: For sequence fragments of five amino acids, there are $20^5 = 3200000$ possible letter combinations. Thus, even for five-residue peptides, certain sequence combinations are massively underrepresented in the training set, which makes it impossible to acquire reliable statistics on their structure.

To alleviate this, we developed so-called reduced amino acid alphabets, where certain amino acids are seen as identical. In this definition, identical sequence fragments can be described as regular expressions. For instance, all aliphatic amino acids ([GAVLI]) or sulfur-containing amino acids ([CM]), or all aromatic amino acids ([YWF]) may be assigned a single identity class. These rules may differ based on the position of the amino acid inside a protein fragment, because for instance, the central amino acid may have a stronger influence on the predicted structure. All reduced amino acid alphabets are available in the Supplementary via http://pbpred.eimb.ru/S/reduced_alphabets.zip.

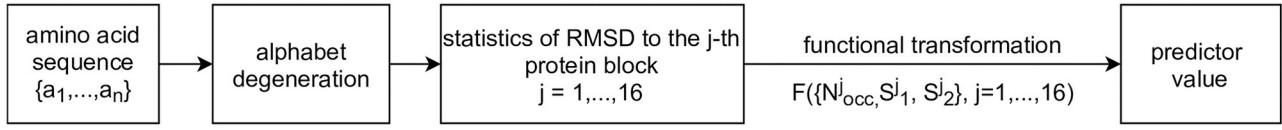

**Fig 3. Generation of predictors based on RMSD statistics between a protein fragment and the 16 basic protein blocks, schematic representation.**

After sequence identity classes are defined, we calculate the minimal sufficient statistics for the distance distribution of each identity class. Namely,

$$\{N_{occ}(seq), \; \mu^j(seq), \; \sigma^j(seq)\}, j = 1, \ldots, 16$$

where $N_{occ}(seq)$ is how many times instances of the sequence $seq$ occur in the training sample, $\mu^j(seq)$ and $\sigma^j(seq)$ are the mean and variance estimates of the RMSD between the fragments from the current identity class and the $j$–th protein block. Further, we applied various non-linear transformations to generate a pool of predictors. Fig 3 schematically represents the predictor generation procedure. One example of an RMSD-based predictor a t-statistics-like function, which captures differences between an identity of a certain sequence class and the whole sample. This and a more detailed description of the generation procedure can be found in S1 File.

**Physico-chemical properties.** Of amino acids were acquired from the AAindex database [26]. Among the 550 different properties present in the database, we used various scales of hydrophobicity, flexibilities of amino acid residues, solvation free energy et.c. We will illustrate predictor generation with an example: Assume, there is a sequence fragment of $n$ residues $\{a_1, \ldots, a_n\}$. Each amino acid $\{a_i, i = 1, \ldots, n\}$ has a certain value of hydrophobicity $\{H_i, i = 1, \ldots, n\}$. Further, a functional transformation was applied to the array of property values $\{H_1, \ldots, H_n\}$, i.e. $F(H_1, \ldots, H_n)$. In the current example, the functional transformation reflects a periodic change of hydrophobicity with period $T$ in a given fragment of $n$ residues:

$$F(H_1, , H_n) = \sqrt{\left(\sum_{k=1}^{n} H_k \cos\left(k\frac{2\pi}{T}\right)\right)^2 + \left(\sum_{k=1}^{n} H_k \sin\left(k\frac{2\pi}{T}\right)\right)^2} \qquad (3)$$

For other physico-chemical properties, functional transformations may be different. Fig 4 illustrates the general procedure for generating this kind of predictors. Note that at this step, no alphabet reduction was used.

## Model fitting

Further, we train a linear regression model to estimate the similarity $D_i(V)$ (1) between a protein fragment $V$ and each of the protein blocks $PB_i$. Assume $\hat{y}$ is the estimate of $D_i(V)$ for some $i = 1, \ldots, 16$, and $\{x_1, \ldots, x_p\}$ are all the predictor values generated above. Then,

$$\hat{y} = \beta_0 + \beta_1 x_1 + \cdots + \beta_p x_p, \qquad (4)$$

where $\beta_0, \ldots, \beta_p$ are the regression coefficients. In order to fit the regression coefficients, we

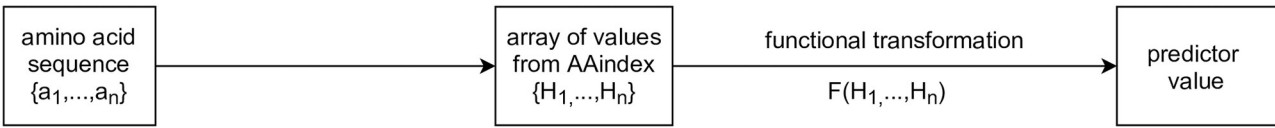

**Fig 4. Generation of predictors based on physico-chemical properties of amino acids comprising a protein fragment, schematic representation.**

adapted the method of stepwise linear regression, with bidirectional elimination of predictors. Model fitting procedure is described in details in S1 File.

## Discussion

The rapidly increasing amount of information about protein structures, coupled with advances in machine learning, opens many possibilities to tackle the problem of protein structure prediction. One of the common limiting factors in local structure prediction is the availability of homologues to the target protein in the PDB. Another challenge is recovering the coordinates of the protein backbone given its sequence.

In this work, we introduced a novel approach to predicting local structure that does not rely on information about protein homologues. It is particularly useful for predicting the local structure of those proteins that do not have any homologues in the PBD. Mainly, we decomposed a complex multidimensional problem of protein stricture prediction into several one-dimensional regression tasks. Each regression problem aims to quantify the relationship between a protein sequence, the statistics of its structural conformations and the physicochemical properties of its amino acids.

Using an appropriate feature space is one of the keys to achieve high prediction accuracy for almost every leaning algorithm. We believe, the main contribution of our method is the generation of a comprehensive pool of features adapted for various challenging scenarios.

In particular, rare sequence fragments posed a significant challenge for training our model. To tackle this, we developed a so-called reduced alphabet: Sequence fragments were mapped to specific regular expressions which took physicochemical properties of the amino acids into account to form new sequence identity classes. This way, we achieved more abundant identity classes with presumably similar conformation properties.

Another set of the features with high predictive power relies on the physical properties of the structural fragments. For instance, predictor (3) reflects the periodic change of hydrophobicity. When the periodicity value $T = 3.6$, it corresponds to a feature with the highest predictive power for alpha-helical conformation (Protein Block 'm'). Thus, when the value of this feature is high, hydrophobic and hydrophilic residues are likely to face opposite direction, which is energetically favourable for a helical fragment on the surface of a protein.

To benchmark our method against the current gold standard, we used the same set of proteins for training and testing as in PB-kPred, and the same set of 16 basic structures (protein blocks). For this dataset, we achieved the Q16 classification accuracy of 67.9% independent of the availability of homologues, which is higher than the 40.8% to 66.3% depending on the availability of homologues by PB-kPred [12].

Nevertheless, the comparison between alignment methods and their performance is not straightforward, and requires to use a range of different criteria [27]. Not only it cannot be captured by the Q16 classification accuracy alone, it is also important to point out that two assignment performed with different distance measures are not exactly the same thing. They may describe very different classes of structures even if the same set of reference PBs or even the same dataset is used.

One major difference between our method and PB-kPred is that RMSD but not RMSDA was used to measure the distance between two protein fragments. Although the calculation of RMSD seems at first computationally more demanding, currently available implementation allowed us circumvent largely expensive computations. We have shown that for local structure prediction, RMSD measure has considerable advantages. Structures that are close in terms of RMSDA may not align well in 3D, while for RMSD it always holds. Further, we have split all 5-residue protein structures into 16 clusters based on the nearest protein block in terms of

RMSD. The clusters that corresponded to regular structures (alpha-helix, beta-sheet) had a high overlap with those based on RMSDA. In contrast, clusters with variable structural composition differed between the two distance measures.

Further, as a proof of concept, we reconstructed the structures of several protein fragments solely from the (actual) RMSD values between a fragment and the 16 protein blocks. Using the notion of degrees of freedom, we established the lowest number of basic structures required to reconstruct the conformation of a protein fragment of a given length, with various geometric constraints. In practice, the reconstruction may benefit from a larger number of basic structures since distances to protein blocks are predicted and contain a certain amount of noise. In future works, we aim to determine the optimal set of basic structures and their effective number via clustering based on RMSD. This way, the protein blocks will be most variable in terms of the same metric as the one used in the regression model.

The small fraction of membrane and NMR-resolved proteins structures in the *PDB30* dataset may not be explained well by our prediction model. Garbuzynskiy et al. show that NMR-resolved structures have systematic differences form the protein structures resolved by X-ray [28]. This batch effect needs to be accounted for if the structures from both methods are to be used together. Additionally, the connection between a protein structure and its physico-chemical properties varies drastically between the membrane and non-membrane proteins. Multi-class prediction performance of our method was indeed higher (72%) for the dataset *PISCES30*, which only contained structures resolved by X-Ray.

The proposed approach can be further extended to several scenarios: Fast PDB mining for similar structures using predicted distances to the basic protein blocks. In molecular dynamics, researchers use predicted structures of small protein fragments as an initial approximation and constraints for the model. To this end, we believe that protein blocks of different length may be needed and utilised together.

## Supporting information

**S1 File.**
(PDF)

**S2 File.**
(TXT)

## Acknowledgments

We thank the anonymous reviewer for the questions and comments that lead to a considerable improvement of the work. We also thank Alexej Viktorovich for the valuable input during the manuscript preparation.

## Author Contributions

**Investigation:** Vladislava Milchevskaya, Alexei M. Nikitin.

**Methodology:** Vladislava Milchevskaya, Ivan V. Filatov, Yury V. Milchevskiy.

**Project administration:** Yury V. Milchevskiy.

**Software:** Alexei M. Nikitin, Sergey A. Lukshin, Yuri V. Kravatsky, Yury V. Milchevskiy.

**Supervision:** Vladimir G. Tumanyan, Natalia G. Esipova, Yury V. Milchevskiy.

**Writing – original draft:** Vladislava Milchevskaya, Yury V. Milchevskiy.

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
