## [Decision Letter · Decision Letter 0]

29 Dec 2020

PONE-D-20-29188

Structural Coordinates: A novel approach to predict protein backbone conformation

PLOS ONE

Dear Dr. Milchevskaya,

Thank you for submitting your manuscript to PLOS ONE. After careful consideration, we feel that it has merit but does not fully meet PLOS ONE’s publication criteria as it currently stands. Therefore, we invite you to submit a revised version of the manuscript that addresses the points raised during the review process.

The reviewer found the manuscript interesting but there is a significant number of issues that must be addressed before the paper can be further considered.

We look forward to receiving your revised manuscript.

Kind regards,

Oscar Millet

Academic Editor

PLOS ONE

Journal Requirements:

"NO"

Reviewers' comments:

Reviewer's Responses to Questions

**Comments to the Author**

1. Is the manuscript technically sound, and do the data support the conclusions?

Reviewer #1: Partly

2. Has the statistical analysis been performed appropriately and rigorously? 

Reviewer #1: I Don't Know

3. Have the authors made all data underlying the findings in their manuscript fully available?

Reviewer #1: No

4. Is the manuscript presented in an intelligible fashion and written in standard English?

Reviewer #1: Yes

5. Review Comments to the Author

Reviewer #1: The work presented by Vladislava Milchevskay takes up a classic concept of description of the protein structure, which starts from the secondary structure to that of the structural alphabet. The authors try to improve on the limitations they have found. The work is quite innovative and interesting, but the article is often too direct and simplistic. As I was interested, I have ask a lot of questions.

In the introduction, a little confusion may arise from reading the definition of the structural alphabet. It is too mixed with the secondary structures. A structural alphabet is normally agnostic and seeks to bring everything together, while secondary structures seek patterns (mainly based on hydrogen bonds), so it is not exactly the same thing. The secondary structures go correctly assigned propellers and sheets, but leave the loops undefined.

It is therefore appropriate to speak of (i) the secondary structures assigned by DSSP & STRIDE, predicted by PSIPRED or SSPRO, then (ii) the structural alphabets.

Several approaches exist to define structural alphabets as described by Calpha (your third paragraph)

and you conclude with the Protein Blocks which are predicted for example by LOCUSTRA.

Likewise, the end of the introduction is a bit fuzzy. The principle of a classification is to put close information together. Here with PBALIGN, iPBA or SAFASTA, structural alphabets have made it possible to find structures that are close, aligned, more or less simply. So, it would be better to bring the questions, the reader has trouble following.

By the way, what does the RMSDA do at the end of the introduction?

Is RMSD based on Calpha, or more atoms?

RMSD and RMSDA have units. They are often forgotten.

I am not sure that 5 numbers after the digits are essential... especially when you know the precision of protein structure resolution.

As results are provided before the Methods, reader had to interpret what the authors wanted to do. What this reviewer had understood: Authors want to use PB (based on dihedral angles) as structural seeds, but used RMSD to do the real assignment. Ok, to be honnest, this reviewer is not entirely sure it is that. So, it must be precise in Results section (and also in Methods section, as it is far to be clear).

From Table 1, we see a clear change in the assignment for at least 1/3 of the pentapeptides. It is an impressive number that must be discuss (i have only sum the difference of frequency, e.g.. 1.662 1.405 3.791 5.836 1.573 0.657 4.868 0.771 2.338 0.905 0.082 0.553 2.722 1.596 0.278 0.082). it must analysed and discussed.

How are assignment exchanges carried out? Do we only observe expected exchanges, for example PB m with PB n, or else surprising ones, for example PB a with PB g? What is the impact of omega angles? Are these only pentapeptides with high RMSDAs? We need an analysis of the consequences of the change in clustering distance. Is it close to the PB similarity distance used in PBALONG or iPBA ?

How was chosen the prototype for Figure 1?

If the distance criteria to say it is improve is the RMSD and the assignment is re-done using RMSD. It is logical that it is better? But how is it in terms of RMSDA? How does it impact mean and sd values?

RMSDA is very quick in terms of CPU times, RMSD is more expensive, isn't it?

Similarly, for PBALIGN, IPBA and myPBA, the PBs were only used as an intermediate before RMSD optimisation. What could be the interest to do RMSDA, then RMDS locally instead of RMSD globally as in the cited researches. The text must be rephrase to focuss on the precise questions of the authors.

What is the percentage of cis/trans omega in protein structures? see the paper about the limited impact of omega angles in protein analyses in Joseph et al Amino Acids 2012 http://www.ncbi.nlm.nih.gov/pubmed/22227866? Most of the differences cis in front of trans are equilibrated at a longer distance as the pentapeptides are overlapping.

There is an interesting discussion on G, that seems not PBg. It could be interesting to discuss it in regards to beta-turns of type IV, i.e. the non-determined turn. Similarly, it must be explain more precisely, why the value of PBg of 1 Ang is so surprising, even if PB m is around 0.2 Ang, i.e. the opposite would have been strange.

"This notion may present a significant challenge for interpretation of the structures that belong to such a non-compact (high variance and large mean RMSD to the central structure) cluster, as well as hamper the prediction accuracy." One missing point is the 'overlapping' effect of PBs, when series of PBs are observed, i.e. PBs mnopac, they are with better approximation, see de Brevern et al Protein Science 2002 paper, is it improve here or not?

How is the true significance of Table 2. In regards to the precision of protein structures, there is no difference for this reviewer. Moreover, a little bit of MDs, and no impact.

The protein dataset is not excellent. If the reviewer understands, it came from Vriend et al. [11], but ref [11] is not the right one. If it had been done with BLASTCLUST, it keeps proteins with identity higher than the chosen threshold. Please proceed to a new evaluation with PISCES datasets that are of really better quality.

I apologize; i have not in the result section understood the evaluation of reconstruction. It is too quick and not detailed.

Similarly for the Q16 section, the prediction approaches presented in Methods section are applied. Provided prediction rates for LOCUSTRA and PB-kPred are the ones from the papers or new evaluations on the specific dataset? not clear and very important.

Finally, the major question. Is it possible to say that it is better or worse? On one side, there is an assignment based on RMSDA, it leads to the assignment of PBs a,b...,o,p; on the other side, there is an assignment based on RMSD with the same number of letters, but a different assignment. How is it possible to compare with Q16(PB-RMSDA) and Q16 (PB-like-RMSD)?

Some important references of Levitt (w/Kolodony) and others are missing (Unger, Fetrow,...). they can provide another view of the question.

Is there an open tool for assignment of these PB-RMSD?

6. PLOS authors have the option to publish the peer review history of their article (what does this mean?). If published, this will include your full peer review and any attached files.

Reviewer #1: No

---

## [Author Response · Author response to Decision Letter 0]

23 Mar 2021

Dear Dr. Millet,

Dear reviewer,

Thank you for giving us the opportunity to submit a revised draft of the manuscript titled 

Structural Coordinates: A novel approach to predict protein backbone conformation to PLOS ONE. We appreciate the time and effort that you and the reviewer have dedicated to providing valuable feedback on our manuscript. We are grateful to the reviewer for their detailed and insightful comments on the paper. We have been able to incorporate changes to reflect most of the suggestions provided by the reviewers. We have highlighted the changes within the manuscript. 

Here is a point-by-point response to the reviewers’ comments and concerns:

 1. In the introduction, a little confusion may arise from reading the definition of the structural alphabet. It is too mixed with the secondary structures. A structural alphabet is normally agnostic and seeks to bring everything together, while secondary structures seek patterns (mainly based on hydrogen bonds), so it is not exactly the same thing. 

The secondary structures go correctly assigned propellers and sheets, but leave the loops undefined. It is therefore appropriate to speak of (i) the secondary structures assigned by DSSP & STRIDE, predicted by PSIPRED or SSPRO, then (ii) the structural alphabets.

Thank you for pointing this out. We agree with this comment, and have changed the introduction to reflect the suggested structure (lines 12-30 of the Revised Manuscript with Track Changes). We have also made minor changes in the remaining text of the Introduction, in order to make it read smooth.

 2. Likewise, the end of the introduction is a bit fuzzy. The principle of a classification is to put close information together. Here with PBALIGN, iPBA or SAFASTA, structural alphabets have made it possible to find structures that are close, aligned, more or less simply. So, it would be better to bring the questions, the reader has trouble following.

Thank you, we agree with this comment. The end of the introduction is re-written in a way to state the question we aim to investigate, hopefully, clearer (lines 44-58 of the Revised Manuscript with Track Changes). Indeed, unlike for PBALIGN and iPBA, our ultimate goal is to be able to reconstruct the 3D coordinates of a protein fragment. Nevertheless, multiclass prediction (i.e. predicting similar structures for a protein fragment knowing its sequence) is an essential step, and we attempt to use it with a slight modification: Namely, by seeing a structural alphabet as “anchors” in the space of structures, and a query fragment being represented in terms of distances to these “anchor” structures. 

Excuse us, we did not manage to find the tool SAFASTA that you have mentioned. We have tried several spelling options, but that unfortunately did not help. We would be happy to incorporate it later if you point us to the tool. 

 3. By the way, what does the RMSDA do at the end of the introduction?

Thank you! We agree, it should not be there. The formula is removed (lines 70-73 of the Revised Manuscript with Track Changes).

 4. Is RMSD based on Calpha, or more atoms?

Thank you for noticing that. We agree, this needs to be clarified. The RMSD is based on N, Calpha, C atoms of the backbone. We have included this clarification in in the text and in the notation description of formula (2) (lines 246, 308, 358 of the Revised Manuscript with Track Changes).

 5. RMSD and RMSDA have units. They are often forgotten.

Thank you for pointing this out, the units are added.

 6. I am not sure that 5 numbers after the digits are essential... especially when you know the precision of protein structure resolution.

Thank you for pointing this out, there is indeed no need for the 5 digits, we have shortened it to one digit in Table 2, and two digits in Table 1.

 7. As results are provided before the Methods, reader had to interpret what the authors wanted to do. What this reviewer had understood: Authors want to use PB (based on dihedral angles) as structural seeds, but used RMSD to do the real assignment. Ok, to be honnest, this reviewer is not entirely sure it is that. So, it must be precise in Results section (and also in Methods section, as it is far to be clear).

Thank you, this is a very valid point, we need to make it clear. Exactly as the reviewer wrote, we used the PBs derived by de Brevern et al as seeds (or cluster centres), but instead of using RMSDA as a similarity measure between a fragment and a seed, we used RMSD. To be able to do so, we reconstructed the 3D coordinates of each PB using the dihedral angles with which they were defined, as well as the standard values for the bond angles, bond lengths and setting the omega value to 180 degrees. To bring more clarity, we emphasised that in the very beginning of the Results subsection (lines 84-90 of the Revised Manuscript with Track Changes). We have also renamed the corresponding subsection of the Results into “Relation of RMSD- and RMSDA-based assignments” (line 75). 

 8. From Table 1, we see a clear change in the assignment for at least 1/3 of the pentapeptides. It is an impressive number that must be discuss (i have only sum the difference of frequency, e.g.. 1.662 1.405 3.791 5.836 1.573 0.657 4.868 0.771 2.338 0.905 0.082 0.553 2.722 1.596 0.278 0.082). it must analysed and discussed.

How are assignment exchanges carried out? Do we only observe expected exchanges, for example PB m with PB n, or else surprising ones, for example PB a with PB g? What is the impact of omega angles? Are these only pentapeptides with high RMSDAs? We need an analysis of the consequences of the change in clustering distance. 

We agree, thank you. There is now a subsection in the Results dedicated to comparing the two assignments (line 75 of the Revised Manuscript with Track Changes), as well as a new Figure (Fig.1). Indeed, we observe more changes in assignments than expected. We believe that the discrepancy between the two distances increases for irregular structures: Such as clusters PB g or PB h. The reason for that may be that the contribution of each dihedral angle to the RMSD is not equivalent (deviation in central angles causes more RMSD-distant fragments), while in the RMSDA there is no distinction between those.

Later the reviewer raised a very valid point about overlapping fragments (Structural Words could be an example), which may compensate for the above. We believe, that is indeed what happens in the case of regular structures, but less so for the irregular ones. We discussed that in details in a newly subsection of the Results, the “Benchmarking using Structural Words”, line 290 of the Revised Manuscript with Track Changes.

 9. Is it close to the PB similarity distance used in PBALONG or iPBA ?

If we understood correctly, the reviewer is referring to GDT_PB (Global Distance Test Total analog for PB alignments) from Gelly et al. (https://doi.org/10.1093/nar/gkr333), which is used to compare alignments derived by different tools. But I think, it is something different from what we do here: We only use a reverse of the distance, i.e. 1/RMSD, as the similarity measure between fragments. Maybe we misunderstood the question.

 10. How was chosen the prototype for Figure 1? (now Figure 2)

Thank you for the question, we have added a clarification in the caption of the Figure 2, page 18 of the Revised Manuscript with Track Changes (former Figure 1) and in the text (lines 169-179 of the Revised Manuscript with Track Changes). The prototype for Fig.2 is just an example that RMSD and RMSDA may have a different meaning. We chose the reference (red) fragment to be an alpha-helix because it is familiar to everyone and is easy to visualise. We then isolated all the fragments with RMSDA=14.1 degrees from the reference fragment, and choose three examples (A-C) to show that the values of RMSD may still vary. Example (D) is designed to show a situation where RMSDA=0 degrees, but the RMSD is rather high. It has mainly an illustrative purpose.

 11. If the distance criteria to say it is improve is the RMSD and the assignment is re-done using RMSD. It is logical that it is better? But how is it in terms of RMSDA? How does it impact mean and sd values?

Thank you for the question. It would have been interesting to explore this aspect. However, 

we mainly plan to use the model presented in the article for 3D coordinate reconstruction. 

For this task, it is better if it produces more defined (compact) classes of structures in terms of RMSD, therefore we did not explore the qualities of our (RMSD-based) assignment in terms of RMSDA further. 

However, we have added a benchmarking case to cover a possibly related question. Using the example of Structural Words (mentioned in the reviewer’s comments later), and there RMSD-based assignment shows an improvement as compared to the RMSDA assignment: The Structural Words generated with RMSD are almost always more compact and the number of words is significantly larger (Table 4 and lines 307-315 of the Revised Manuscript with Track Changes).

 12. RMSDA is very quick in terms of CPU times, RMSD is more expensive, isn't it?

Thank you for the question. We ourselves were surprised to discover that the computing time for both RMSD and RMSDA is comparable. The reason for this is that in case of RMSDA, in order to find a dihedral angle, one needs to solve for two planes and then compute an angle between the corresponding normal vectors. For a fragment of 5 residues, there are 8 dihedral angles, which means we repeat the above procedure 8 times. On the other hand, for the RMSD, the alignment procedure requires computation of eigenvectors and eigenvalues for a 3x3 matrix, followed by the change of coordinates, but it is done once for the complete fragment. This is a commonly used algorithm, therefore there are very efficient implementations of it. Hence, the use of RMSD did not hinder the computational time. 

 13. Similarly, for PBALIGN, IPBA and myPBA, the PBs were only used as an intermediate before RMSD optimisation. What could be the interest to do RMSDA, then RMDS locally instead of RMSD globally as in the cited researches. The text must be rephrase to focuss on the precise questions of the authors.

Thank you for pointing this out. We agree that the previous text might have been misleading. We have added two paragraphs explaining why we chose RMSD over RMSDA (§s 184-197 of the Revised Manuscript with Track Changes), and that we needed a particular quality of the chosen distance (lines 219-222 of the Revised Manuscript with Track Changes): Namely, that small distance value guarantees well-aligned structures. 

But I thought you might be interested in the underlying idea, and wrote a longer answer.

Our main idea was to find a way to retain structural information about a protein fragment after transition to the structural alphabet. Therefore, we used distanced to the PBs as new “structural” coordinates. This approach is not yet fully developed, but as a proof of concept we show that indeed a protein conformation (in terms of dihedral angles) can be reconstructed from its 16 structural coordinates. In this context, one could potentially substitute RMSD with RMSDA, and still reconstruct the conformation (this is a guess). But further on we predict RMSDs to the Protein Blocks based on the sequence (not only assign the fragment to a certain class). For this, we use physicochemical properties of individual amino acids, and it becomes crucial that small values of the distance correspond only to closely aligned structures. Fig. 2 shows that this is not always the case for RMSDA, and without this quality the training of our prediction model cannot really work. 

We have not included this complete explanation to the text of the article (it is probably too wage to be in a paper), but it is a very interesting question. We would take your advice and add it, if it is reasonable.

 14. What is the percentage of cis/trans omega in protein structures? see the paper about the limited impact of omega angles in protein analyses in Joseph et al Amino Acids 2012 http://www.ncbi.nlm.nih.gov/pubmed/22227866? Most of the differences cis in front of trans are equilibrated at a longer distance as the pentapeptides are overlapping.

 15. There is an interesting discussion on G, that seems not PBg. It could be interesting to discuss it in regards to beta-turnxs of type IV, i.e. the non-determined turn. Similarly, it must be explain more precisely, why the value of PBg of 1 Ang is so surprising, even if PB m is around 0.2 Ang, i.e. the opposite would have been strange.

"This notion may present a significant challenge for interpretation of the structures that belong to such a non-compact (high variance and large mean RMSD to the central structure) cluster, as well as hamper the prediction accuracy." One missing point is the 'overlapping' effect of PBs, when series of PBs are observed, i.e. PBs mnopac, they are with better approximation, see de Brevern et al Protein Science 2002 paper, is it improve here or not?

Thank you for this suggestion. This is a great benchmarking example that we have overlooked before. In the revised version, we have assessed the noted paper to explore the effect of overlapping fragments and have added a subsection in the manuscript (line 290 of the Revised Manuscript with Track Changes). 

 16. How is the true significance of Table 2. In regards to the precision of protein structures, there is no difference for this reviewer. Moreover, a little bit of MDs, and no impact.

Thank you for pointing this out, we agree that original text was not detailed enough. The

difference between a standard MD and our approach is the loss function we aim to optimise. Usually, some sort of energy function is optimised to recover the protein conformation. We, however, do following: For a proposed protein conformation,

one can calculate the RMSDs to the 16 Protein Blocks, which is a function of dihedral angles. The Loss function we aim to optimise is the mean squared error between the two “coordinates” of a protein conformation in the space of our structural alphabet. Table 2 shows that we can reconstruct the true protein conformation from the “structural coordinates” with high precision. 

We have updated the subsection “Reconstruction of the backbone conformation” to make the difference between the MD and our approach clearer (lines 198-228, but in particular lines 211-223 of the Revised Manuscript with Track Changes).

 17. The protein dataset is not excellent. If the reviewer understands, it came from Vriend et al. [11], but ref [11] is not the right one. If it had been done with BLASTCLUST, it keeps proteins with identity higher than the chosen threshold. Please proceed to a new evaluation with PISCES datasets that are of really better quality.

Thank you for spotting the mistake in the reference, we corrected that.

We also agree that the PISCES dataset suggested by the reviewer is much better. We used it to assess the performance of our method. Since there are no transmembrane proteins (PISCES only contains X-Ray structures) in the new data set, and they differ from the non-membrane proteins in their physicochemical properties significantly, our multi class prediction increased to 72%.

For benchmarking against PB-kPRED, we still used the old data set (PDB30) since it is used by the authors.

 18. I apologize; i have not in the result section understood the evaluation of reconstruction. It is too quick and not detailed.

Thank you for pointing this out. We agree with the comment and changed the text of the subsection to make it clearer (lines 198-228).

 19. Similarly for the Q16 section, the prediction approaches presented in Methods section are applied. Provided prediction rates for LOCUSTRA and PB-kPred are the ones from the papers or new evaluations on the specific dataset? not clear and very important.

Thank you, we agree that this is a very important point. We have changed the text of the subsection and clearly stated that the dataset for LOCUSTRA is different from PDB30 (lines 270-273, Revised Manuscript with Track Changes). 

 20. Finally, the major question. Is it possible to say that it is better or worse? On one side, there is an assignment based on RMSDA, it leads to the assignment of PBs a,b...,o,p; on the other side, there is an assignment based on RMSD with the same number of letters, but a different assignment. How is it possible to compare with Q16(PB-RMSDA) and Q16 (PB-like-RMSD)?

Thank you for raising this point. We agree that the meaning of the RMSDA assignment is different from that of the RMSD, and it would be incorrect to compare the assignments per se. We have added the following line to the discussion: “Nevertheless, the comparison between alignment methods and their performance is not straightforward, and requires to use a range of different criteria. Not only it cannot be captured by the Q16 classification accuracy alone, it is also important to point out that two assignment performed with different distance measures are not exactly the same thing. They may describe very different classes of structures even if the same set of reference PBs or even the same dataset is used.” (lines 449-453)

However, the Q16 problem is defined in each case: We know the true assignment of the structure by the chosen metric, i.e. which PB is the closest. Then we aim to predict the closest PB (in the same metric) from the sequence of a protein fragment. Assignments themselves are not comparable, but the frequency with which we make a correct prediction is. In the end, either assignment is an approximation of the original structure, and the desire is to make it correct (and meaningful) as often as possible.

One could argue that it is possible to construct another set of 16 classes such that they are uninformative but the prediction accuracy it higher. We believe, it is not the case here, but, as the reviewer rightly noted, it is important to point the difference of the two assignments.

 21. Some important references of Levitt (w/Kolodony) and others are missing (Unger, Fetrow,...). they can provide another view of the question.

Thank you! This was a helpful suggestion, and made us aware of certain complications in comparing of structural alignment methods, which we overlooked before (lines 449-453). We have added the Kolodony reference (27). 

 22. Is there an open tool for assignment of these PB-RMSD?

Thank you for asking. We agree it needs to be made available. We have added the tool to the homepage of the server: https://pbpred.eimb.ru/S/index.html (“Program PB-RMSD”, second line).

In addition to the above comments, we corrected the spelling mistakes we have notice during the editing process.

We look forward to hearing from you in due time regarding our submission. We will ge glad respond to any further questions and comments you may have.

Sincerely,

Vladislava Milchevskaya

on behalf of all the authors

---

## [Decision Letter · Decision Letter 1]

15 Apr 2021

Structural Coordinates: A novel approach to predict protein backbone conformation

PONE-D-20-29188R1

Dear Dr. Milchevskaya,

We’re pleased to inform you that your manuscript has been judged scientifically suitable for publication and will be formally accepted for publication once it meets all outstanding technical requirements.

Kind regards,

Oscar Millet

Academic Editor

PLOS ONE

Additional Editor Comments (optional):

Reviewers' comments:

Reviewer's Responses to Questions

**Comments to the Author**

1. If the authors have adequately addressed your comments raised in a previous round of review and you feel that this manuscript is now acceptable for publication, you may indicate that here to bypass the “Comments to the Author” section, enter your conflict of interest statement in the “Confidential to Editor” section, and submit your "Accept" recommendation.

Reviewer #1: All comments have been addressed

2. Is the manuscript technically sound, and do the data support the conclusions?

Reviewer #1: Yes

3. Has the statistical analysis been performed appropriately and rigorously? 

Reviewer #1: Yes

4. Have the authors made all data underlying the findings in their manuscript fully available?

Reviewer #1: Yes

5. Is the manuscript presented in an intelligible fashion and written in standard English?

Reviewer #1: Yes

6. Review Comments to the Author

Reviewer #1: I am particularly pleased and happy with the authors' responses to my many suggestions. I find that the paper has been greatly improved, and is also much more rigorous.

I would have just noted that DSSP assigns 8 states, and STRIDE 7 (only one type of bend). I apologise for the bad writing, it was SA-FAST (https://link.springer.com/article/10.1186/1471-2105-9-349).

7. PLOS authors have the option to publish the peer review history of their article (what does this mean?). If published, this will include your full peer review and any attached files.

Reviewer #1: No

---

## [Editor Report · Acceptance letter]

22 Apr 2021

PONE-D-20-29188R1 

Structural Coordinates: A novel approach to predict protein backbone conformation  

Dear Dr. Milchevskaya:

I'm pleased to inform you that your manuscript has been deemed suitable for publication in PLOS ONE. Congratulations! Your manuscript is now with our production department. 

Kind regards, 

on behalf of

Dr. Oscar Millet 

Academic Editor

PLOS ONE